# Engineered Human Monoclonal scFv to Receptor Binding Domain of *Ebolavirus*

**DOI:** 10.3390/vaccines9050457

**Published:** 2021-05-04

**Authors:** Jaslan Densumite, Siratcha Phanthong, Watee Seesuay, Nitat Sookrung, Urai Chaisri, Wanpen Chaicumpa

**Affiliations:** 1Graduate Program in Immunology, Department of Immunology, Faculty of Medicine Siriraj Hospital, Mahidol University, Bangkok 10700, Thailand; jaslan.den@student.mahidol.ac.th (J.D.); siratcha.pha@student.mahidol.edu (S.P.); watee.see@student.mahidol.ac.th (W.S.); 2Center of Research Excellence on Therapeutic Proteins and Antibody Engineering, Department of Parasitology, Faculty of Medicine Siriraj Hospital, Mahidol University, Bangkok 10700, Thailand; 3Biomedical Research Incubation Unit, Department of Research, Faculty of Medicine Siriraj Hospital, Mahidol University, Bangkok 10700, Thailand; nitat.soo@mahidol.ac.th; 4Department of Tropical Pathology, Faculty of Topical Medicine, Mahidol University, Bangkok 10400, Thailand; urai.cha@mahidol.ac.th

**Keywords:** human single-chain antibodies (HuscFvs), cell-penetrating antibodies, *Ebolavirus*, *Ebolavirus*-like particles, Glycoprotein (GP), Cleaved GP (GPcl), Thermolysin, receptor-binding domain (RBD), receptor binding site (RBS), Niemann-Pick C1

## Abstract

(1) Background: *Ebolavirus* (EBOV) poses as a significant threat for human health by frequently causing epidemics of the highly contagious Ebola virus disease (EVD). EBOV glycoprotein (GP), as a sole surface glycoprotein, needs to be cleaved in endosomes to fully expose a receptor-binding domain (RBD) containing a receptor-binding site (RBS) for receptor binding and genome entry into cytoplasm for replication. RBDs are highly conserved among EBOV species, so they are an attractive target for broadly effective anti-EBOV drug development. (2) Methods: Phage display technology was used as a tool to isolate human single-chain antibodies (HuscFv) that bind to recombinant RBDs from a human scFv (HuscFv) phage display library. The RBD-bound HuscFvs were fused with cell-penetrating peptide (CPP), and cell-penetrating antibodies (transbodies) were made, produced from the phage-infected *E. coli* clones and characterized. (3) Results: Among the HuscFvs obtained from phage-infected *E. coli* clones, HuscFvs of three clones, HuscFv4, HuscFv11, and HuscFv14, the non-cell-penetrable or cell-penetrable HuscFv4 effectively neutralized cellular entry of EBOV-like particles (VLPs). While all HuscFvs were found to bind cleaved GP (GPcl), their presumptive binding sites were markedly different, as determined by molecular docking. (4) Conclusions: The HuscFv4 could be a promising therapeutic agent against EBOV infection.

## 1. Introduction

*Ebolavirus* (EBOV) is a highly contagious pathogen causing severe illness with rapid progression and high mortality rates, i.e., *Ebolavirus* disease (EVD) or Ebola hemorrhagic fever (EHF), which is endemic in the African territory [1]. EBOV is a filamentous, enveloped, non-segmented negative-sense RNA virus (about 14,000 nm in length with a diameter of 80 nm) that belongs to the genus *Ebolavirus* of the family *Filoviridae*, which also include Genera *Cuevavirus*, *Dianlovirus*, and *Marburgvirus* (MARV) [2,3]. The EBOV RNA genome is about 18–19 kb in size and encodes seven proteins, including nucleoprotein (NP), which encases the genomic RNA; virion protein (VP) 35, which has polymerase co-factor activity and the ability to suppress the host’s innate immunity for immune evasion; VP40, which drives the progeny virus assembly and budding; glycoprotein (GP), which functions in host cell attachment and virus entry; transcription factor VP30, which forms complex with the L (polymerase) protein for protein synthesis and genome replication; VP24, which can inhibit interferon signaling; and L protein, which is the viral RNA-dependent RNA polymerase [4]. To date, six species of EBOV have been identified, including *Zaire ebolavirus*, *Sudan ebolavirus*, *Tai Forest ebolavirus*, *Bundibugyo ebolavirus*, *Reston ebolavirus*, and *Bombali ebolavirus* [5]. The six species differ in the disease severity that they cause; the *Zaire ebolavirus* causes the most severe form of EVD, while the *Reston ebolavirus* causes EVD in non-human primates and has not been known to cause human disease [6].

EBOV expresses the glycoprotein (GP) on the virion surface. During the viral replication and assembly, the GP is produced, cleaved post-translationally by the host enzyme, furin yielding two disulfide-linked GP1 and GP2 subunits [7]. The GP1 which facilitates host cell attachment and receptor binding for cellular entry, consists of a glycan cap (GC), a heavily-glycosylated mucin-like domain (MLD), and a receptor-binding domain (RBD) containing a putative receptor binding site (RBS) [8]. The GP2 mediates fusion of host endosomal and viral membranes for the virus genome release into cytosol for further replication. The GP2 consists of two heptad repeat regions and the internal fusion loop (IFL). The crystal structure of GP shows that the protein exists as a bowl-like trimeric GP1/2, in which the GP1-GP2 form the base of the bowl and the bulky carbohydrates of GC and MLD form the bowl head, shielding the underneath RBD [8]. Upon attachment with several host attachment factors [9,10,11,12,13], the virus is internalized by macropinocytosis into endosome [14,15]. Endosomal cathepsins in late endosome cleave the GP at residues 190–194, removing the whole of GC and MLD, leaving the 19-kDa GP1 disulfide-linked to GP2, called GPcl [16,17,18]. Cleavage of the GP by cathepsins exposes the RBS in the RBD, which binds to domain C of the authentic receptor, Nieman-Pick C1 (NPC1), expressed in the endosome [19]. Binding of the GPcl to the NPC1 induces conformation changes of the GP, releasing the IFL of the GP2, and mediates viral-host membrane fusion, which is followed by the virus genome uncoating and cytosol exit [20].

Since GP is present on the EBOV surface, it is the main target to develop therapeutic interventions and prophylactic vaccines against EBOV infection and EVD. A vast number of monoclonal antibodies (mAbs) targeting different epitopes of GP have been developed successfully, such as ZMapp, ZMAb, or MB-003 [21,22,23], but neither of these mAbs were shown to recognize RBD, which is highly conserved among filoviruses [8]. It is rational to assume that any substance that interferes with the RBD function should be a broadly effective anti-filovirus agent. However, only a few RBD-targeting mAbs with weak-to-moderate neutralizing activity have been identified [24,25,26]. In this study, engineered human single-chain antibody variable fragments (HuscFvs) that bind to RBD and interfere with the cellular entry of the EBOV-like particles were generated. The fully human antibodies could be developed further towards the clinical application, either for passive prophylaxis or treatment of the EBOV infection.

## 2. Materials and Methods

### 2.1. Culture Media, Cells, Antibodies, and Reagents Used in This Study

Dulbecco’s modified Eagle’s medium (DMEM), Opti-MEM (a reduced-serum medium, which is an improved Minimal Essential Medium (MEM)), fetal bovine serum (FBS), and recombinant full-length Zaire EBOV GP were from ThermoFisher Scientific, Waltham, MA, USA. GlutaGro (the stabilized dipeptide form of l-glutamine) and Penicillin-Streptomycin were from Corning, NY, USA.

Human embryonic kidney 293T (HEK293T) cells and HeLa cells were from ATCC, Manassas, VA, USA. They were cultured in DMEM and supplemented with 10% FBS, 2 mM GlutaGro, and 1× Penicillin-Streptomycin (complete medium) at 37 °C in humidified 5% CO_2_ incubator.

Mouse monoclonal anti-6× His antibodies was from Bio-Rad, Hercules, CA, USA. Rabbit poly-clonal anti-E-tag antibodies and horseradish peroxidase (HRP)-conjugated VeriBlot IP detection reagent were from Abcam, Cambridge, UK. Alkaline phosphatase (AP)-conjugated goat anti-mouse IgG and AP-conjugated goat anti-rabbit IgG were from SouthernBiotech, Birmingham, AL, USA. Horseradish peroxidase (HRP)-conjugated mouse IgGκ-BP was from Santa Cruz Biotechnology, Dallas, TX, USA. Rabbit polyclonal anti-GP antibodies, Alexa-Fluor 488/567-conjugated goat anti-mouse IgG (H+L), and Alexa-Fluor 567-conjugated goat anti-rabbit IgG (H+L) were from Invitrogen, Carlsbad, CA, USA. Thermolysin, a thermostable neutral metalloendopeptidase enzyme (non-specific peptidase) produced by the Gram-positive bacteria, *Bacillus thermoproteolyticus*, was from Sigma-Aldrich, St. Louis, MO, USA.

### 2.2. Plasmid Constructs

The plasmids constructs, including pET21a^+^-RBD (coding for residues 54–201 of EBOV GP), pCI-Neo GP1/2, and pLVX-Puro GP1/2-IRES-VP40-Bla (bicistronic vector coding for EBOV GP and VP40 N-terminally fused with β-lactamase; bla), were commercially synthesized by GenScript, Piscataway, NJ, USA.

### 2.3. Production of Recombinant EBOV Receptor-Binding Domain (RBD)

To produce recombinant EBOV RBD, BL21 (DE3) *Escherihia coli* carrying pET21a^+^-RBD (residues 54–201 of EBOV GP) (GenScript, Picataway, NJ, USA) was grown to mid-log phase (OD600 nm 0.4–0.6) in Luria-Bertani (LB) broth and supplemented with 100 µg/mL of ampicillin (LB-A broth). Protein expression was induced by adding isopropyl b-D-1-thiogalactopyranoside (IPTG) (Thermo Fisher Scientific) (1 mM final concentration) to the bacterial culture, and the bacteria were incubated for an additional 5 h. The recombinant protein in the bacterial inclusion bodies was purified, as described previously [27]. Briefly, bacterial cells were lysed with BugBuster protein extraction reagent (Merck, Darmstadt, Germany) and supplemented with Lysonase bioprocessing reagent (Merck). The bacterial homogenate was centrifuged, and the inclusion bodies in the sediment was washed sequentially with a series of wash buffers. The inclusion bodies was solubilized with 50 mM CAPS, pH 11, 0.3% sarkosyl, and 1 mM dithiotreitol (DTT). The solubilized proteins (1 mg/mL) was refolded by dialysis twice against 20 mM imidazole, pH 8.0, and 0.1 mM DTT at 4 °C with slow stirring and then two more times against 20 mM imidazole and pH 8.0 without DTT. The preparation was filtered through a 0.2-μm low protein binding Acro-disc^®^ Syringe Filter (Pall, Port Washington, NY, USA) and kept in a 30 °C water bath for an additional 2 h. Protein concentration of the preparation was measured (Pierce^®^ BCA Protein Assay), while purity was checked by SDS-PAGE and Coomassie Brilliant Blue G-250 (CBB) staining. The preparation was added with trehalose (60 mM final concentration) and kept in small portions at 4 °C (or at −80 °C for long-term storage).

### 2.4. Production of Human Single-Chain Antibody Variable Fragments (HuscFvs) That Bind to the EBOV RBD

Phage display technology was used for production of the HuscFvs that bind to the EBOV RBD, as described previously [27,28]. Briefly, a well of a microplate was coated with 1 µg of purified recombinant EBOV RBD in phosphate buffered saline, pH 7.4 (PBS). After blocking the unoccupied sites of the well surface with 5% bovine serum albumin (BSA) in PBS, 100 µL of HuscFv phage display library [28] was added to incubate with the immobilized RBD. The antigen-unbound phages were removed by washing; the antigen-bound phages in the well were added with mid-log phase-grown HB2151 *E. coli* and incubated at 37 °C for 3 h to allow phage infection. The phage infected-*E. coli* were spread onto selective agar plates (2× YT agar containing 100 µg/mL of ampicillin; 2× YT-A). The bacterial colonies that appeared on the plates after overnight culture at 37 °C were screened for the presence of the HuscFv genes (*huscfvs*) by colony PCR using the pCANTAB 5E phagemid specific primers [28]. *E. coli* clones carrying *huscfv*-phagemids were grown in 2× YT-A broth and 2% glucose (2× YT-AG) to the mid-log phase, then IPTG (1 mM final concentration) was added to each *E. coli* culture. Expression of soluble E-tagged-HuscFvs and binding of the HuscFvs in the *E. coli* lysates to the RBD were screened by indirect ELISA and Western blotting. The *huscfvs* carried by the positive RBD-binding clones were subjected to DNA sequencing. The DNA sequences were deduced into polypeptide sequences. Complementarity determining regions (CDRs) and their respective canonical immunoglobulin framework regions (FRs) of the sequenced HuscFv polypeptides were predicted using an online International ImMunoGeneTics (IMGT^®^) Information System.

### 2.5. Large Scale Production of the HuscFvs and Generation of Cell-Penetrating HuscFvs

For production of cell-penetrating HuscFvs (transbodies) and large-scale production of the transbodies and the original HuscFvs, the *huscfvs* in the pCANTAB 5E phagemids were subcloned into pLATE52 expression vector (Thermo Fisher Scientific) with and without DNA sequence coding for cell-penetrating peptide, i.e., nona-arginine (R9) [29], by using the ligation independent cloning (LIC) method (aLICator LIC Cloning and Expression Kit 4; Thermo Fisher Scientific), as described previously [27]. The *R9-huscfv*/*huscfv-*pLATE52 expression vectors were introduced into JM109 *E. coli*, and the recombinant vector-positive clones were cultured; the plasmids were extracted and sequenced. The verified plasmids were introduced separately into Rosetta (DE3) *E. coli.* Transformed Rosetta (DE3) *E. coli* colonies were grown individually in 250 mL of 2× YT broth, containing 100 mg/mL ampicillin, 34 mg/mL chloramphenicol, and 1 mM IPTG. The R9-HuscFvs/HuscFvs were purified from the bacterial inclusion bodies as for the recombinant RBD and were determined by SDS-PAGE and CBB staining. Binding of the R9-HuscFvs and HuscFvs to the recombinant RBD was rechecked by indirect ELISA and Western blotting.

### 2.6. Western Blotting

Protein samples were separated by SDS-PAGE (either 10% or 14% gel) and electrotransblotted onto nitrocellulose membranes. The blotted membranes were blocked with 5% skim milk for 2 h. The blocked membranes were washed, probed with primary antibodies overnight, washed three times with buffer, i.e., 20 mM Tris, 150 mM NaCl, and 0.1% Tween-20 (TBS-T), and probed with secondary antibodies for 1 h. Membranes were developed with Luminata Forte Western HRP substrate (Merck, NJ, USA), and the antigen-antibody reactive bands were visualized by using the ImageQuant LAS 4000 imager.

### 2.7. Biocompatibility of R9-HuscFvs/HuscFvs to Mammalian Cells

HEK293T cells grown in 96-wells-plate were treated with indicated concentration of the R9-HuscFvs/HuscFvs or medium alone (non-toxic control) for 24 h. Cell cytotoxicity was determined, as described previously [27], by measuring protease released from dead cells into culture medium by using Cytotox-Glo cytotoxicity assay (Promega, WI, USA).

### 2.8. Determination of Cell-Penetrating Ability of R9-HuscFvs

HeLa cells were seeded on cover slips and treated overnight with 30 µg/mL of R9-HuscFvs. The cells were washed, fixed with 4% paraformaldehyde in PBS for 30 min, permeabilized with 0.1% Triton-X 100 in PBS for 15 min, and blocked with 5% BSA in PBS for 1 h. The cells were then stained with anti-6× His antibody for 1 h, followed by fluorophore-conjugated secondary antibody (1:300 dilution) at 4 °C for 1 h, and counterstained with DAPI (1:5000 dilution) at 4 °C for 15 min. Localization of the R9-HuscFvs in the stained cells was visualized by 1-µm sectional confocal microscopy.

### 2.9. Preparation of EBOV-Like Particles and Transmission Electron Microscopy

HEK293T cells established in the 100-mm culture dish were transfected with 25 µg of pLVX-Puro GP_1/2_-IRES-VP40-bla by using Xfect transfection reagent (Clontech, Mountain View, CA, USA). Culture medium was discarded post-transfection. The cells were washed twice with PBS, replenished with Opti-MEM, and incubated at 37 °C in humidified 5% CO_2_ atmosphere. Culture supernatant containing Ebola VLP-bla was collected at 72 h post-transfection and centrifuged at 4000× *g* for 10 min to remove cell debris, and the supernatant containing the VLPs was kept in small aliquots at −80 °C until use.

For visualization of the VLP by transmission electron microscopy (TEM), copper grids (300 mesh) were covered with formvar film and 50 μL of the VLP samples were applied onto the grids for 10 sec. The grids were then washed 3 times with distilled water and negatively stained with 2% (*w*/*v*) uranyl acetate for 1 min. The stained grids were dried in a desiccator and observed at 80 keV by using Hitachi model HT7700 transmission electron microscope.

### 2.10. Determination of the Ability of the R9-HuscFvs/HuscFvs to EBOV RBD in Inhibiting Cellular Entry of the VLP

For testing ability of the R9-HuscFvs/HuscFvs in inhibiting the VLP cellular entry, HEK293T cells (3 × 10^4^ cells) in 100 µL complete DMEM were seeded into individual wells of 96-wells culture plate and kept in 37 °C incubator with 5% CO_2_ atmosphere overnight. VLP-containing supernatant (100 µL) were pre-incubated with R9-HuscFvs/HuscFvs (6 µg) or diluent (served as mock treatment) at 37 °C for 30 min. The antibody-VLP mixtures or boiled VLP alone (served as mock infection, since EBOV-GP is denatured by heating) were added into wells containing the HEK293T cells. VLP-containing supernatant was also added into wells containing culture medium alone (background control). The plate was then spinoculated by centrifugation at 2000× *g* at room temperature for 45 min and then incubated at 37 °C in a humidified 5% CO_2_ incubator for 3 h. The β-lactamase activity of the VLP-bla in the cells was detected by using GeneBLAzer in vivo detection kit (ThermoFisher Scientific), according to manufacturer’s instructions. Differential VLP entry of different treatment groups was analyzed, by measuring blue (Em460) and green (Em530) fluorescence intensity, and visualized under UV light by using an inverted fluorescence microscope. Calculation of the parameters were as follows:Blue/Green ratio = (Em460 of test − EM460 of background) ÷ (EM530 of test − Em530 of background)(1)
Percent VLP entry = (Blue/Green ratio of test) ÷ (Blue/Green ratio of mock treatment) × 100(2)
Percent neutralization = 100 − Percent VLP entry(3)

### 2.11. Preparation of Cells That Exposed EBOV GP-RBD

For preparing cells that exposed EBOV GP-RBD, thermolysin was used to treat the GP-expressed cells [16]. Firstly, HeLa cells that overexpress EBOV-GP were prepared: HeLa cells (4 × 10^5^ cells in 2 mL complete DMEM) in individual wells of the six-wells-culture plate were transfected with 5 µg of pCI-Neo GP_1/2_ by using the Xfect transfection reagent. After transfection, the medium was replaced with complete DMEM and the cells were cultured for 48 h. For preparation of HeLa cells that exposed EBOV GP-RBD on the surface, the GP-expressed HeLa cells were treated with thermolysin. Thermolysin (Sigma-Aldrich) was reconstituted to 5 mg/mL in HNC buffer (20 mM HEPES, 150 mM NaCl, and 2 mM CaCl_2_) and kept at −80 °C as thermolysin stock. HeLa cells expressing EBOV GP or non-transfected HeLa cells (served as control) were washed twice with PBS, gently added with thermolysin solution (250 µg in 1 mL HNC buffer), and incubated at 37 °C in the humidified CO_2_ incubator for 1 h. The reaction was stopped by adding EDTA to the final concentration of 10 mM and washed the cells twice with complete DMEM and once with PBS.

### 2.12. Determination of Binding of R9-HuscFvs/HuscFvs to RBD-Exposed Cells

HeLa cells expressing GP or HeLa cells expressing thermolysin-treated GP (RBD-exposed cells) were seeded on cover slips. The cells were treated with 30 µg/mL of R9-HuscFvs/HuscFvs and incubated at 37 °C for 2 h to allow cell adherence. The cells were then washed, fixed with 4% paraformaldehyde in PBS for 30 min, blocked with 5% BSA in PBS for 1 h, and stained with anti-6× His (1:200 dilution) and anti-GP antibodies (1:500 dilution) for 1 h. The cells were then stained with fluorophore-conjugated anti-isotype antibody (1:300 dilution) for 1 h and counterstained with DAPI (1:5000 dilution) for 15 min at 4 °C. The stained cells were mounted onto slides, sealed with nail polish, and subjected to sectional confocal microscopy by using Carl Zeiss LSM 700 laser scanning.

For flow cytometric analysis of the binding of R9-HuscFvs/HuscFvs to the RBD, HeLa cells expressing GP or HeLa cells were treated with thermolysin, as described above. The cells were incubated with 1 µg/mL of the antibodies in FACS buffer (2% FBS and 0.02% sodium azide in PBS) on ice for 1 h. The cells were washed and stained with anti-6× His (1:200 dilution) at room temperature for 1 h, followed by the fluorophore-conjugated secondary antibody (1:300 dilution) on ice for 1 h. The stained cells were fixed with 1% paraformaldehyde in PBS and analyzed by using the BD LSR Fortessa flow cytometer.

### 2.13. Protein Modeling and Molecular Docking

Homology models of the three-dimensional (3D) structures of HuscFvs were obtained by submitting the respective HuscFv amino acid sequences to the Iterative Threading Assembly Refinement (I-TASSER) service [30]. ModRefiner [31] was then used to refine the HuscFvs models to improve the physical quality of the structures by making amino acid backbone side-chain atoms to be completely flexible. Subsequently, Fragment-Guided Molecular Dynamics (FG-MD) [32] was used to make the primarily modeled structures closer to their native structures, and the local geometry of the structures was improved by removing the steric clashes and improving the torsion angle and the hydrogen-binding networks. The ClusPro 2.0 server [33] was used for intermolecular docking between HuscFv models and the available crystal structures of EBOV trimeric GPcl [34] (PDB: 5F1B) or monomeric GPcl [19] (PDB: 5HJ3). PROtein binDIng enerGY prediction (PRODIGY) [35] was used to determine the target-antibody interaction with the lowest energy. The intermolecular bonds between target-antibody complexes were identified by the Protein Interactions Calculator (PIC) server [36]. PyMOL software (The PyMOL Molecular Graphics System, Version 1.3r1 edu, Schrodinger, New York, NY, USA) was used to visualize the modeled complexes, and the HuscFv residues that presumptively formed contact interface with the targets were identified.

### 2.14. Statistical Analysis

Data obtained from independent experiments were analyzed by using one-way ANOVA or an unpaired *t*-test with a *p* value < 0.05 considered as significantly different.

## 3. Results

### 3.1. Recombinant EBOV GP-RBD and HuscFvs to the RBD

To isolate HuscFvs that bound to RBD, we firstly produced and purified recombinant RBD (rRBD) from transformed *E. coli.* The recombinant protein was overexpressed in inclusion bodies, which was subsequently purified non-chromatographically and refolded, as described previously [27]. Figure 1A,B shows SDS-PAGE-separated and Western blot patterns of the recombinant protein after purification. The recombinant protein was finally verified as the EBOV RBD by liquid chromatography-tandem mass spectrometry (LC-MS/MS) (data not shown).

Next, we used the purified rRBD as a bait antigen to isolate HuscFv-displayed phages that bound to rRBD by phage biopanning [28]. Purified rRBD was immobilized onto the surface of a well, and the HuscFv phage display library [28] was subsequently added. The rRBD-bound HuscFv-displayed phages were rescued [28]. Screening for the presence of HuscFv genes (*huscfv*) in phage-transformed *E. coli* clones by colony PCR [28] showed 12 *huscfv*-positive clones (Figure 1C). Soluble HuscFvs were produced from the *huscfv*-positive *E. coli* clones. Western blotting showed that 10 of 12 *huscfv*-positive clones could express adequate HuscFvs (30–35 kDa) (Figure 1D). HuscFvs of 4 out of 10 *E. coli* clones (clones 1, 4, 11, and 14) bound to rRBD, as shown by indirect ELISA and Western blotting (Figure 1E,F). The *huscfvs* of the rRBD-bound HuscFv clones were subjected to DNA sequencing.

### 3.2. Cell-Penetrating HuscFvs and Their Biocompatibility to Human Cells

Since the target RBD on the EBOV particle, including VLP, is not exposed until internalized and the GP is cleaved by cathepsins in late endosomes [17], the HuscFvs should be cell-penetrable to interfere with the RBD bioactivity. Cell-penetrating peptides (CPP) are short peptide sequences and, when fused, are capable to translocate the fused partner through plasma membranes. Several CPPs have been identified with different characteristics or origin, such as *Drosophila*-derived penetratin [37], HIV-derived Tat [38], polyarginine [39], etc. In this study, we made use of nona-arginine (R9), which is relatively short, small in size, and has low immunogenicity. The *huscfvs* of the *E. coli* clones that expressed HuscFvs bound to rRBD were subcloned from phagemids into the pLATE52 plasmid construct coding for N-terminal R9 [27]. The R9-HuscFvs were produced on a large-scale as inclusion bodies, purified non-chromatographically, and refolded as for rRBD. The purity of R9-HuscFvs was high, as shown by SDS-PAGE and CBB staining, and Western blotting (Figure 2A,B). Binding ability of the R9-HuscFvs were assessed by ELISA and Western blotting, which revealed that the R9-HuscFvs were able to bind rRBD as the original HuscFv counterparts, but not to recombinant GP (Figure 2C,D)

Cytotoxicity of R9-HuscFvs to human cells was tested using HEK293T cells as a representative. HEK293T cells were treated overnight with 30 μg/mL of individual R9-HuscFvs. Cellular cytotoxicity caused by R9-HuscFvs was analyzed by measuring proteases released from dead cells. We found that R9-HuscFvs of *E. coli* clones 4, 11, and 14 (R9-HuscFv4, R9-HuscFv11, and R9-HuscFv14) did not cause measurable cytotoxicity, whereas R9-HuscFv1 was highly toxic and caused about 90% total cell death (Figure 2E). From these results, the R9-HuscFv1 was not tested further.

Next, we tested the cell-penetrating ability of R9-HuscFvs. HeLa cells were treated overnight with individual R9-HuscFvs at 30 μg/mL, and intracellular R9-HuscFvs were stained and analyzed by confocal microscopy. As expected, all R9-HuscFvs were able to translocate and retained intracellularly (Figure 2F and Appendix A).

### 3.3. Ability of the R9-HuscFvs/HuscFvs in Inhibiting Cellular Entry of the Ebola VLP

Determination of antiviral activities of therapeutic candidates, including antibodies, is inevitably the best when studying with the native, authentic viruses. However, EBOV, categorized as the risk group 4 pathogen, requires biosafety level-4 facilities, which are not available worldwide. Virus-like particles (VLP), which are devoid of viral genome and therefore non-replicative, are widely used in place of the authentic viruses in several study aspects, such as virus biology, development of vaccines, and therapeutic antibodies. We used the Ebola VLP-based reporter system, previously established elegantly by Tscherne et al. [40]. The VLPs consist of EBOV GP and VP40 matrix protein that are necessary for VLP assembly and budding. In this study, the VP40 was fused with β-lactamase enzyme as a reporter protein, which can be detected intracellularly by cell-permeable fluorescence resonance energy transfer (FRET) substrate CCF2-AM. Cleavage of the substrate by β-lactamase in VLP-infected cells resulted in cells fluorescing blue, whereas the un-cleaved substrate in uninfected cells resulted in cells’ fluorescing green (Figure 3A). VLPs were produced and purified by density-gradient ultracentrifugation. Characterization of purified VLPs showed the presence of GP by Western blotting (Appendix A). Analysis of VLP by TEM revealed its morphology in filamentous shape about 80 nm in diameter (Appendix A), corresponding with EBOV morphology [41]. The VLPs were preincubated with individual R9-HuscFvs, and the VLP-antibody mixtures were used to spinoculate HEK293T cells. We found that the R9-HuscFv4 significantly reduced the number of VLP-infected cells, while the R9-HuscFv11 and R9-HuscFv14 could not (Figure 3B,C).

Given that RBD is exclusively exposed intracellularly, following cathepsin cleavage, we wondered whether the CPP is required for HuscFv inhibitory activity. To answer this, the non-cell penetrable HuscFvs were produced, purified, and confirmed for rRBD-binding activity (Appendix A). Surprisingly, it was found that the HuscFv4 could reduce the VLP cellular entry to the same extent of the R9-HuscFv4 (Figure 3D,E). Taken together, we have shown that HuscFv4, whether cell-penetrable or non-cell-penetrable, could inhibit cellular entry of the Ebola-VLP.

### 3.4. Binding of the R9-HuscFvs/HuscFvs to RBD

Thermolysin is a protease that cleaves GP at physiological pH, yielding the cleaved GP (GPcl, the 19-kDa GP1 linked to GP2), similar to endosomal cathepsins [18]. To demonstrate RBD-HuscFv binding, GP-expressed HeLa or HeLa cells were treated with thermolysin before incubating with HuscFvs. Binding of HuscFvs on the cell surface was analyzed by confocal microscopy and flow cytometry. Non-cell-penetrable HuscFvs were used to demonstrate the RBD binding, since cell-penetrable R9-HuscFvs generated higher background staining (data not shown), possibly by their cell-penetrating ability. We found that HuscFv4 co-localized with GPcl (Figure 4A). Analysis by flow cytometry showed that the HuscFv4 bound to thermolysin-treated GP-expressed cells but not to thermolysin-treated GP-lacking cells (Figure 4B). Unexpectedly, the non-neutralizing HuscFv11 and HuscFv14 also co-localized with GPcl and also bound to thermolysin-treated GP-expressed cells (Figure 4A,B). Since only the HuscFv4 was capable of neutralizing the VLP cellular entry (Figure 3B–F), these data suggest that probably the HuscFv4, HuscFv11, and HuscFv14, bound at different RBD sites, approached the target GPcl via different angles or mediated different target occupancy.

### 3.5. Computerized Simulation of HuscFv-RBD Interaction

To investigate presumptive binding characteristics of the HuscFvs, deduced amino acid sequences of individual HuscFvs were subjected to computerized homology modelling and presumptive binding sites were obtained by docking the HuscFv-three dimensional (3D) models with the crystal structure of monomeric GPcl (PDB: 5F1B) by Wang et al. [19] or trimeric GPcl (PDB: 5HJ3) by Bornholdt et al. [34]. Consensus models revealed that HuscFv4 and HuscFv14 bound to trimeric GPcl, whereas HuscFv11 bound to the monomeric GPcl (Figure 5A). Although the target sites of the HuscFv4, HuscFv11, and HuscFv14 are located similarly and mainly within the GP1 core and only a few residues of the GP2 (Figure 5A), the HuscFvs accessed the target site via different approaching angles. The HuscFvs bound to the GPcl protomer at the site, which is usually occupied by the glycan cap (GC) in the un-cleaved GP [8] (Appendix A), which explains the non-reactivity of the HuscFvs to the un-cleaved GP. It should be noted that, while the HuscFv11 bound to the medial surface of the apex of monomeric GPcl (Figure 5 and Figure 6B), the HuscFv4 and HuscFv14 bound at the interface of GP1 core and GP2 helix, located about 80 and 90 degrees laterally from the center of GPcl trimer, respectively (Figure 5A). As expected, all HuscFvs mainly use a complementary determining region (CDRs) of heavy and light chain (hereinafter referred as CDRH and CDRL) to interact with the GPcl (Figure 5A and Figure 6A–C; the CDRH and CDRL are shown in dark and light purple loops, respectively). Analysis of GPcl residues interacting with HuscFvs (Figure 6) revealed one major difference between HuscFv4 and HuscFv14, while HuscFv14-interacting residues were exclusively clustered at GP1 core of one GPcl protomer and the distal part of the GP2 fusion loop (Figure 5B, right panel, and Figure 6C), and residues of the GPcl that formed contact with HuscFv4 were “discontinuous” and scattered to S90 and G149 of adjacent GPcl protomers, as well as the GP2 fusion loop (Figure 5B, left panel, and Figure 6A).

## 4. Discussion

Human single-chain antibodies (HuscFvs) that bind to EBOV recombinant RBD (rRBD) were generated by using phage display technology, as described previously [27,28]. The HuscFvs were engineered to be cell-penetrable by molecular linking to a cell-penetrating peptide, nona-arigine (R9). R9-HuscFvs of one rRBD-bound phage transformed-*E. coli* clone, R9-HuscFv4, significantly reduced Ebola VLP cellular entry. RBD is known to be highly conserved among Ebolaviruses, so it is a promising target for anti-EBOV drug development. Structurally, RBD (or GP1 core) contains a receptor binding site (RBS), which forms a sea wave-like structure, with a rising crest and a recessed trough [26] (Appendix A). The RBS crest and trough consist of hydrophilic charged- and hydrophobic amino acids, respectively. In EBOV, the RBS trough is occluded by glycan cap, making it unexposed, while the crest is partially exposed (Appendix A). Numerous anti-EBOV monoclonal antibodies (mAbs) have been identified, with different epitope-specificities and degrees of neutralizing activity. Only a handful of mAbs (intact four-chain IgG) obtained from B cells of human *Marburgvirus* (MARV) survivors that targeted MARV RBS were reported [25]. However, these mAbs did not neutralize EBOV, since MARV RBS is structurally more exposed than that of EBOV RBS and antibodies targeting RBS and/or RBD in natural EBOV infection are rarely described. To our knowledge, only one report described intact IgG mAb from the EVD survivor that bound to the tip of the RBS crest of EBOV in the un-cleaved GP [42]. Based on the crystal structure of the un-cleaved GP in complex with KZ52 mAb solved by Lee et al. [8], the RBS of the un-cleaved GP is covered with glycan cap (GC) diagonally about 45 degrees to the side (Appendix A, grey). The presence of GC has no effect on KZ52 binding, since this mAb binds to the conformational epitope at the base of the trimeric GP (Appendix A). Due to the orientation of GC, antibodies that target RBD and/or RBS are likely to approach from the top or diagonally to the side in the vicinity of the GC. In this study, we have shown that the generated HuscFvs from *huscfv*-phage-transformed *E. coli* clones bind to thermolysin-treated GP (Figure 4B), but not to full-length recombinant GP and membrane-associated GP, as shown by ELISA and confocal microscopic analysis (data not shown). Computerized homology modeling of the HuscFvs and molecular docking suggested that all of the generated HuscFvs approached the RBD target from the top or diagonally in the vicinity of GC (Figure 5A).

Analysis of deduced amino acid sequences of HuscFvs revealed that the CDRH3 of HuscFv4 is exceptionally longer than those of HuscFv11 and HuscFv14 (17 vs. 13 and 10 residues). This exceptionally long CDRH3 of HuscFv4 might explain the neutralizing activity of HuscFv4; the antibody might protrude its CDRH3 into the RBS trough or other clefts, hence interfering with the interaction of RBD with NPC1. Docking results of HuscFv4 revealed the residue S103 of CDRH3, protruded into the cleft, and interacted with G131 of the GP1 core between two GPcl protomers (Figure 5 and Figure 6A). The presumptive binding of the HuscFv4 to the GP1-GP2 interface might explain its neutralizing activity, since this region has been shown to be vulnerable epitopes of the EBOV GP [43], even though the HuscFv4 contact surface did not involve RBS (Figure 5B). It is worth noting that interfering RBS-NPC1 interaction does not seem to require exclusively direct binding of the inhibitor to RBS. Misasi et al. [44] showed that base-binding antibodies, KZ52, and mAb100 could interfere with GPcl binding to NPC1 domain C. Antibody binding to GPcl regions outside the RBS, such as at the base or core of GP1, could make the GP structure more rigid; thus preventing the GPcl conformation change, which consequently interferes with the GPcl-NPC1 interaction [19]. This should explain the ability of the HuscFv4 in inhibiting the VLP cellular entry. However, experiments to demonstrate the HuscFv4 epitope, such as the use X-ray crystallography, to analyze antigen-antibody complex or phage mimotope search and phage peptide alignment with the target sequence, and competitive assay using the phage peptide as the binding competitor [45] are needed to locate the actual epitope.

Even though the HuscFv14 also bound to GPcl, it did not neutralize the VLP cellular entry. This might be due to the differences in pattern of GPcl residues interacting with HuscFv4 and HuscFv14 (Figure 5B and Figure 6A,C). The HuscFv4 bound to more scattered regions of GPcl of two adjacent GP protomers and the G131 in the cleft, while the binding sites of the HuscFv14 is limited to one GP protomer. Although both HuscFvs used the CDRH3 to interact with the GPcl, the CDRH3 of the HuscFv14 was much shorter than that of the HuscFv4. Analysis of cell-penetrating ability of R9-HuscFvs revealed that the R9-HuscFv11 and R9-HuscFv14 were detected in high intensity and distributed almost evenly in cytosol, while the R9-HuscFv4 was weakly detected as several intracellular specks (Appendix A), possibly trapped in endosomes [46]. Because GP cleavage and receptor binding occur exclusively in endosomes, endosomal entrapment might be advantageous for R9-HuscFv4 to bind to and interfere with GPcl-mediated receptor binding and cytoplasmic entry. The inability of HuscFv11 in inhibiting the cellular entry of the VLP should be due to inability to bind functionally active GPcl trimer.

Unexpectedly, we found that the non-cell-penetrable HuscFv4 also reduced VLP cellular entry, similar to the cell-penetrable counterpart. This was unexpected as RBS is occluded until the GC and MLD are cleaved-off; the non-cell-penetrable HuscFv4 could not bind extracellularly. Intensive macropinocytosis has been reported to mediate uptake of albumin and albumin-conjugated recombinant proteins [47,48]. Because EBOV employs macropinocytosis as a major route of cell entry [14,15], it is possible that HuscFvs might be coincidentally up-taken, together with Ebola VLP into endosomes, at which the HuscFvs could exert their bioactivity. The HuscFv phage display library used in this study was constructed from circulating B cells by using degenerate oligonucleotide primers to amplify the immunoglobulin variable genes in order to expand the HuscFv diversity of the phage library [28]. The HuscFvs derived from phages of this library occasionally yielded polyspecific, as we have reported [49]. It is noteworthy that EBOV utilizes various attachment receptors to facilitate entry (although NPC1 is the only entry receptor) [9,10,11,12,13]. HuscFv4 might be polyspecific, which cross-reacts with these attachment receptors and consequently interferes with the VLP cellular entry at the attachment step. Experiments are needed to verify the speculation.

Glycosylation is known to contribute to the antigenicity of particular proteins, and hence glycosylated antigens are generally better used for the development of therapeutic antibodies. However, using non-glycosylated bacterial-derived RBD in this study as the target of HuscFvs is reasonable since RBD has not been shown to contain any predicted glycosylations [50].

Since scFv is about 5 times-smaller in size than that of the intact IgG, it would offer more target accessibility (high penetrating ability), especially the densely packed structure, such as EBOV GP trimer. Replacement of intact IgG by scFv would theoretically increase target-binding occupancy, thereby increasing neutralizing potency. Additionally, one factor that determines the protective efficacy of antibodies in vivo and must be taken into consideration is the antibody-dependent enhancement (ADE) phenomenon, which has been reported in EBOV infection [51]. ADE caused by anti-EBOV antibodies was shown to be unrelated to epitopes [51]. Incomplete viral neutralization is usually caused by poorly or moderately neutralizing antibodies at sub-neutralizing concentration, which influences ADE [51]. Since HuscFv4 lacks Fc portion and is a fully human protein, the antibody fragments should be safe for use in passive immunization and treatment of the EBOV infection and EVD cases.

## 5. Conclusions

With phage display technology, we have isolated, produced, and characterized human monoclonal scFvs (HuscFvs) that bind to recombinant RBD, a conserved moiety of Ebolaviruses. Engineered human monoclonal scFv derived from one of the *E. coli* clones infected with HuscFv-displayed phages, i.e., HuscFv4, was found to neutralize VLP cellular entry, regardless of the cell-penetrating peptide (CPP), suggesting possible alternative mechanism(s) of inhibition of VLP cellular entry, besides interfering with the interaction with the entry receptor. Given that RBD is highly conserved, prediction of presumptive binding sites, together with data reported by others, suggested that binding to RBD does not always correlate with neutralizing activity, as several factors are needed to be considered, such as the antibody’s actual binding site, antibody orientation (angle of target approaching), or target accessibility/binding occupancy. The HuscFv4 generated in this study has a potential as a therapeutic for pre-exposure prophylactic or post-exposure treatment against *Ebolavirus*.

## Figures and Tables

**Figure 1 vaccines-09-00457-f001:**
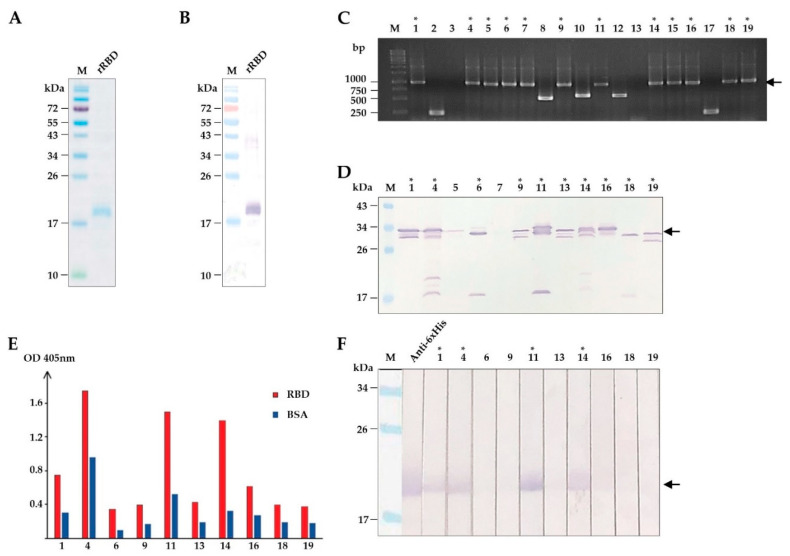
Production of recombinant (r) RBD and selection of HuscFvs that bind to rRBD. (**A**) SDS-PAGE-separated and CBB stained purified rRBD. (**B**) Western blot pattern of rRBD that corresponds to lane rRBD in (**A**) detected by anti-6X His antibody. (**C**) Screening of phage-infected *E. coli* clones that contained rRBD-bound phage clones for *huscfvs* by PCR. Arrowhead indicates the expected amplicon of intact *huscfvs* about 1000 base pairs (bp). (**D**) Screening of soluble HuscFvs contained in the lysates of *E. coli* carrying *huscfv*-phagemids by Western blotting. Asterisks indicate the clones that produced HuscFvs. Arrowhead indicates the expected molecular mass of HuscFvs about 30–35 kDa. (**E**) Binding assay of soluble HuscFvs in *E. coli* lysates to rRBD by indirect ELISA, using BSA as the control antigen. (**F**) Binding assay of soluble HuscFvs in *E. coli* lysates to rRBD blotted onto nitrocellulose strips. Asterisks indicate the clones that their HuscFvs bound to rRBD. Arrowhead indicates the rRBD-HuscFv reactive bands. Lane Anti-6× His was rRBD probed with anti-6X His antibody that served as a positive control for rRBD binding. The number on each lane in (**C**,**D**,**F**) indicates the number of the phage-transformed *E. coli* clone. The asterisks in (**C**,**D**,**F**) indicate HuscFv-displayed phage-infected *E. coli* clones that carried intact *huscfvs*, expressed soluble HuscFvs, and produced rRBD-bound HuscFvs, respectively. Lanes M and numbers on the left of (**A**,**B**,**D**,**F**) are protein molecular mass marker and protein molecular weights in kDa, respectively. Lane M and numbers on the left of (**C**) are DNA ladder and DNA sizes in bp, respectively.

**Figure 2 vaccines-09-00457-f002:**
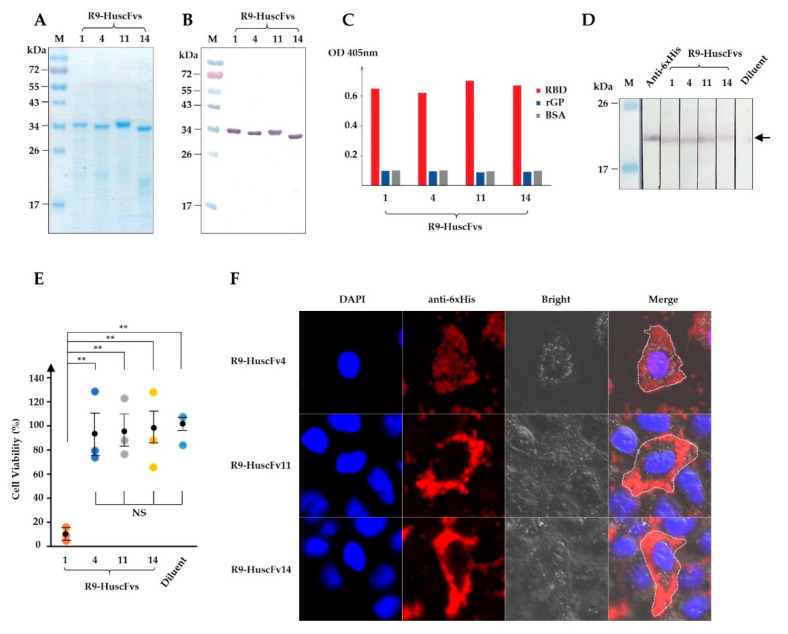
Production of R9-HuscFvs and their rRBD binding activity, biocompatibility to human cells, and cell-penetrating ability. (**A**) R9-HuscFvs were produced, purified, and verified by SDS-PAGE and CBB staining. (**B**) Western blotting of R9-HuscFvs that corresponds to (**A**), detected by anti-6× His antibody. (**C**) Binding of R9-HuscFvs to rRBD and recombinant full-length GP (rGP) by indirect ELISA. (**D**) R9-HuscFvs binding activity to rRBD blotted onto nitrocellulose strips. Arrowhead indicates the rRBD-R9-HuscFvs reactive bands. (**E**) Analysis of R9-HuscFv-mediated cellular cytotoxicity to HEK293T cells incubated overnight with 30 μg/mL of R9-HuscFvs or equivalent volume of antibody diluent. Results are shown as mean ± standard error of percent cell viability of three independent experiments in duplicates. **, *p* < 0.01; NS, not significantly different. (**F**) Cell-penetrating ability (intracellular localization) of R9-HuscFvs in HEK293T cells incubated with 30 µg/mL of R9-HuscFvs for 24 h, as observed by confocal microscopy. Each image shown is a representative of a 1-µm section from a series of z-stack analysis. Lanes M and numbers on the left of (**A**,**B**,**D**) are the protein molecular weight marker and protein molecular masses in kDa, respectively.

**Figure 3 vaccines-09-00457-f003:**
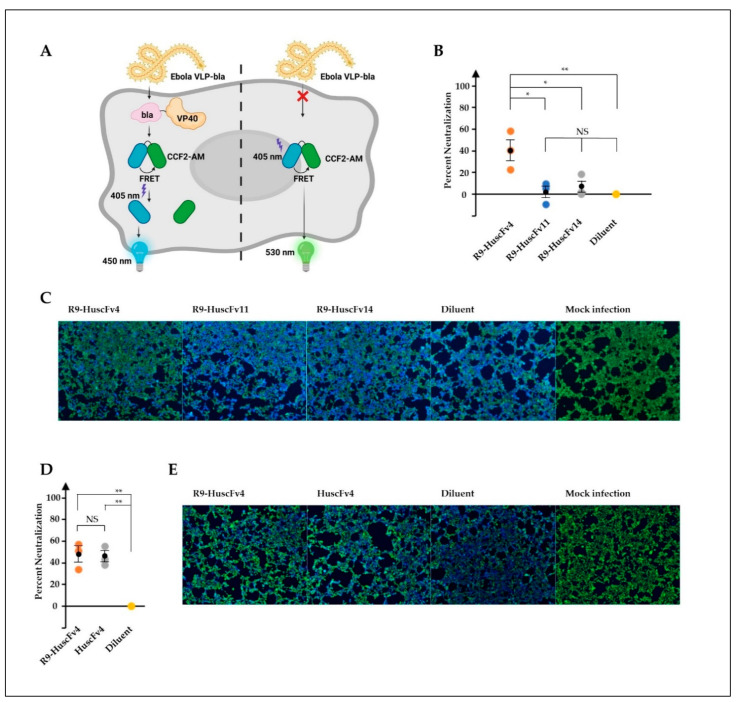
Ability of R9-HuscFvs in inhibition of VLP cellular entry (neutralizing activity). (**A**) Ebola-VLP-based system used in this study. VLP consists of EBOV GP on the surface and VP40 matrix proteins fused with beta-lactamase (bla), which was produced by co-expression of EBOV GP and VP40-bla. Entry of VLP into cells were monitored by measuring bla activity, which cleaved cell-permeable fluorescence resonance energy transfer (FRET) substrate CCF2-AM (green color, emission wavelength at 530 nm), yielding the product (blue color, emission wavelength at 460 nm). (**B**–**E**) HEK293T cells were spinoculated with VLP, pre-incubated with indicated R9-HuscFvs (30 μg/mL final concentration in wells), the equivalent volume of diluent (diluent), or boiled VLP (mock infection). Percent neutralization (inhibition of cellular entry) was calculated as described in Materials and Methods and is shown in (**B**,**D**). Representative images of experiments in (**B**,**D**) are shown in (**C**,**E**), respectively. Results in (**B**,**D**) are shown as mean ± standard error of three independent experiments in triplicates. *, *p* < 0.05; **, *p* < 0.01; NS, not significantly different.

**Figure 4 vaccines-09-00457-f004:**
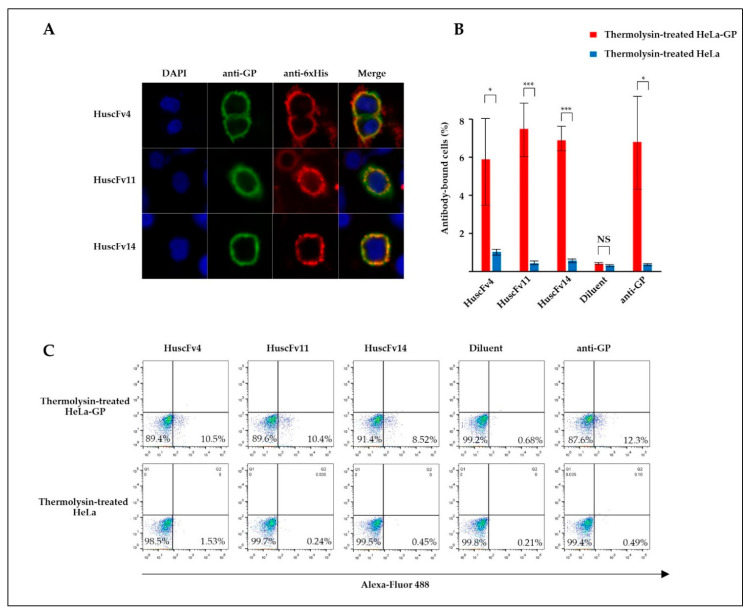
Binding of HuscFvs to GPcl. (**A**) GP-expressed HeLa cells (HeLa-GP) were treated with thermolysin at 37 °C for 1 h. Cells were washed and seeded on cover slips in the presence of 30 μg/mL of indicated HuscFvs at 37 °C for 2 h, followed by staining with primary (anti-6× His and anti-GP antibodies) and fluorophore-conjugated secondary antibodies. Co-localization of GPcl (green) and HuscFvs (red) was visualized by confocal microscopy (orange/yellow in Merge). (**B**) Percent HuscFv-binding to HeLa-GP or HeLa cells. HeLa-GP or HeLa cells were treated with thermolysin, as described in (**A**), and incubated with 1 µg/mL of indicated HuscFvs or diluent on ice for 1 h, followed by staining with primary anti-6× His or anti-GP antibodies and Alexa-Fluor 488-conjugated secondary antibodies. Stained cells were analyzed by flow cytometry. Results are shown as mean ± standard error of three independent experiments. (**C**) The result and gating strategy of one representative experiment in (**B**) are shown. *, *p* < 0.05; ***, *p* < 0.001; NS, not significantly different.

**Figure 5 vaccines-09-00457-f005:**
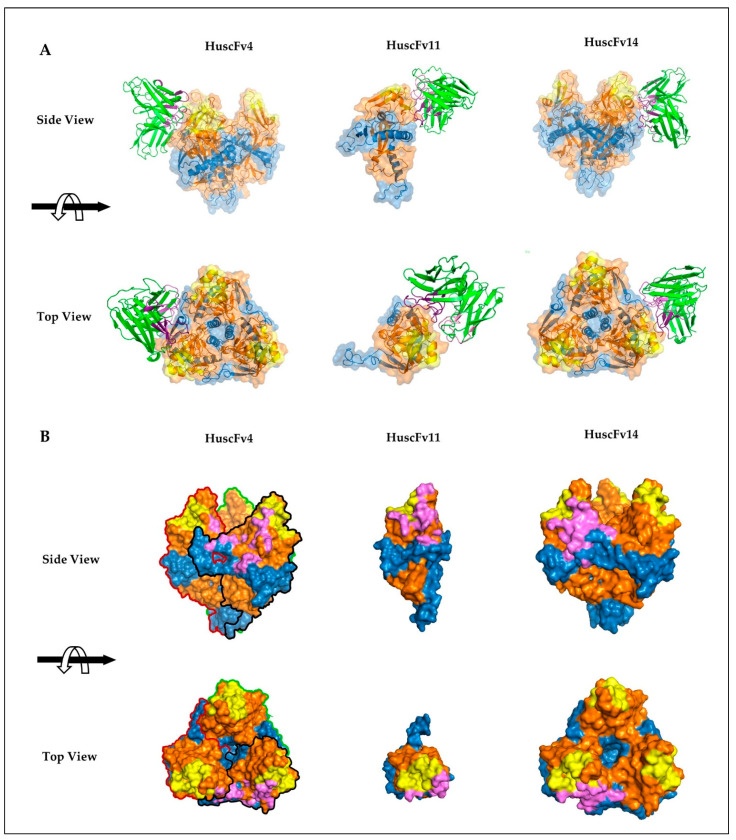
Presumptive binding of HuscFvs to GPcl predicted by computerized homology modeling and intermolecular docking. (**A**) Interactions of HuscFv4 and HuscFv14 to trimeric GPcl (PDB: 5F1B) (left and right panels, respectively) and HuscFv11 to monomeric GPcl (PDB: 5HJ3) (middle panel). The HuscFv-GPcl complexes are shown as the side view (upper panel) or top view (lower panel). The HuscFvs are colored in green, in which CDRH and CDRL loops are colored as dark and light purple, respectively. GP1, GP2, and RBS of the GPcl are in orange, blue, and yellow, respectively. (**B**) Footprints in HuscFv-GPcl complexes as described in (**A**). Contact surface areas between the HuscFvs and the GPcl are colored in purple. The GP1, GP2, and RBS of each GPcl protomer are colored as described in (**A**). Solid black, red, and green lines indicate boundaries of individual GPcl protomers.

**Figure 6 vaccines-09-00457-f006:**
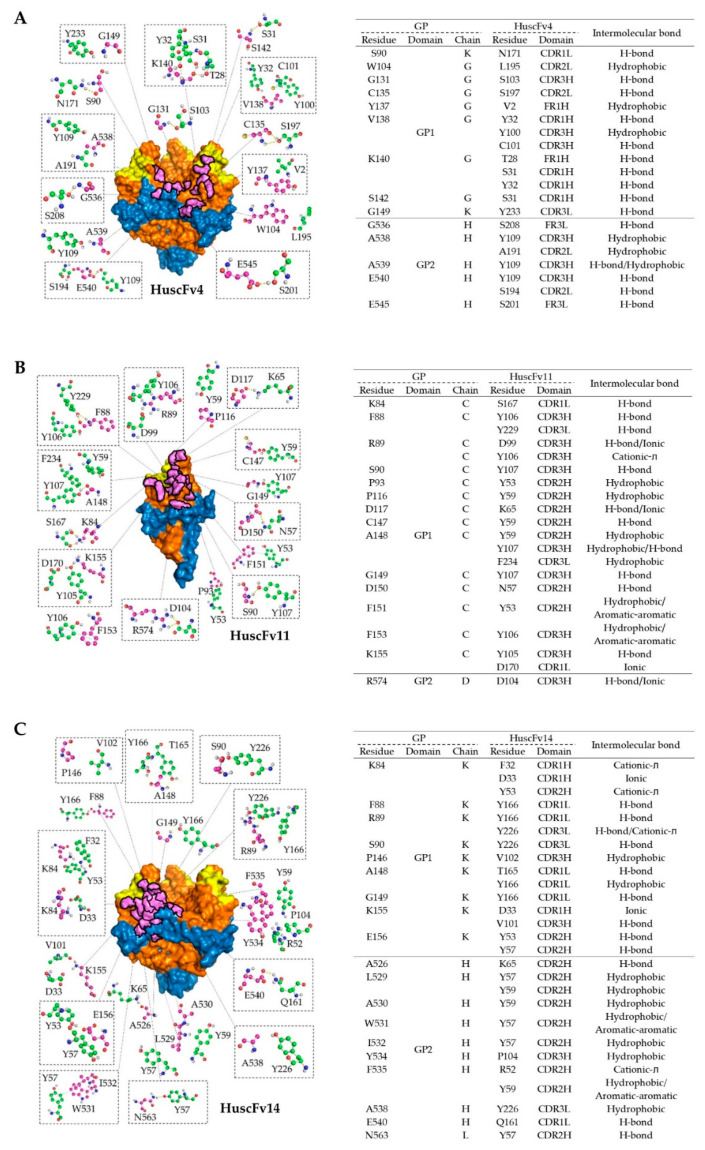
Intermolecular docking-guided presumptive interaction of GPcl residues with HuscFvs. (**A**–**C**) Ball-and-stick models showing presumptive interaction of HuscFv4 (**A**), HuscFv11 (**B**), and HuscFv14 (**C**), with residues of trimeric GPcl or monomeric GPcl as determined by computerized homology modeling and intermolecular docking, as described in Figure 5B. The amino acid backbones and side chains of GPcl (purple) and HuscFvs (green) are shown. GP1, GP2, and RBS of the GPcl and contact surface areas between the HuscFvs and the GPcl are colored as described in Figure 5B. Tables at the right of each respective figures indicate the detailed residues and domains of GPcl and HuscFvs that formed the contact interface.

## Data Availability

All datasets presented in this study are included in the article/Appendix A.

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
