# Peer review of "Engineered Human Monoclonal scFv to Receptor Binding Domain of Ebolavirus"

_vaccines, 2021, doi:10.3390/vaccines9050457_

Round 1
Reviewer 1 Report
In this study, single chain antibodies against the receptor binding domain (RBD) of evolavirus GP1 protein were prepared by the phage display method and examined in terms of their affinities and neutralizing activities. All the three antibodies obtained had an affinity to RBD, while only one of them showed a neutralizing activity probably due to the differences in their binding region and orientation. To further analyze their differences, the authors carried out a docking simulation in silico. This work was well conducted and resulted in preparing desired antibodies successfully. However, it is curious why the epitope analysis was carried out only in silico? Actually, molecular modeling has made a remarkable development in the past few decades, However, experimental support is, in my opinion, still required. Actually, some people even use X-ray crystallography to analyze antigen-antibody complex. Therefore, in my opinion, it would be necessary to include some experimental results in order to discuss about the epitopes of the antibodies.
Reviewer 2 Report
Dear authors of the manuscript:
Engineered Human Monoclonal scFv to Receptor Binding Domain of Ebolavirus,
Thank you for the nice article describing the generation of three anti-GP-RBD human scFvs, identified by phage display technology, utilizing the recombinant RBD.
Overall I would like to suggest to accept this manuscript for publication.
I just have two additional questions:
1) Could you comment about the importance of GP and RBD glycosylation in correspondence to your your identified clones , Fv4, Fv11 & Fv14? as you have used recombinant RBD from a bacterial expression system and for the inhibition of cellular entry you have used VLP produced in Hek293T cells.
2) You have seen no difference for the R9-HuscFv4 and HuscFv4 in VLP neutralization.
Do you have any data that suggest this is due to preincubation of the VLP together with the HuscFv4 and the following the pinocytotic uptake of both to the endosome? Would it be different if you incubate both components separately to the HEK293T cells and wash the cells in between?
Is it possible for the HuscFv4 to attach to the VLP before internalization to the cells?
Could you comment in the discussion on this two questions?
Kind regards
Round 2
Reviewer 1 Report
I understand the author's response, but I still feel that in this paper it is discussed too much about the structure without data showing how these docking models resemble the real structure. So, the title of Figure 6 should be written so that readers can easily realize that these are pictures of docking model.
Author Response
Thank you very much for your worthy points and comments. We have edited the Legend of Figure 6 as suggested, in order to make it clear, i.e., being the in silico results.